# Paracrine rescue of MYR1-deficient *Toxoplasma gondii* mutants reveals limitations of pooled *in vivo* CRISPR screens

**Francesca Torelli**[1,2†], **Diogo M da Fonseca**[1,2†], **Simon W Butterworth**[1,3], **Joanna C Young**[1,4], **Moritz Treeck**[1,2]*

[1]Signalling in Apicomplexan Parasites Laboratory, The Francis Crick Institute, London, United Kingdom; [2]Host-Pathogen Interactions Laboratory, Gulbenkian Institute for Molecular Medicine, Oeiras, Portugal; [3]Whitehead Institute, Massachusetts Institute of Technology, Cambridge, United States; [4]Institute of Immunology and Infection Research, University of Edinburgh, Edinburgh, United Kingdom

**\*For correspondence:**
moritz.treeck@gimm.pt

†These authors contributed equally to this work

## eLife Assessment

This **important** study shows that *Toxoplasma gondii* uses paracrine mechanisms, in addition to cell-intrinsic methods, to evade the host immune system, with MYR1 playing a key role in transporting effector molecules into host cells. The authors present **convincing** evidence that in vivo, MYR1-deficient parasites can be rescued by wild-type parasites, revealing a limitation in pooled CRISPR screens, where such paracrine effects may obscure the identification of key parasite pathways involved in immune evasion

**Abstract** *Toxoplasma gondii* is an intracellular parasite that subverts host cell functions via secreted virulence factors. Up to 70% of parasite-controlled changes in the host transcriptome rely on the MYR1 protein, which is required for the translocation of secreted proteins into the host cell. Mice infected with MYR1 knock-out (KO) strains survive infection, supporting a paramount function of MYR1-dependent secreted proteins in *Toxoplasma* virulence and proliferation. However, we have previously shown that MYR1 mutants have no growth defect in pooled *in vivo* CRISPR-Cas9 screens in mice, suggesting that the presence of parasites that are wild-type at the *myr1* locus in pooled screens can rescue the phenotype. Here, we demonstrate that MYR1 is not required for the survival in IFN-γ-activated murine macrophages, and that parasites lacking MYR1 are able to expand during the onset of infection. While ΔMYR1 parasites have restricted growth in single-strain murine infections, we show that the phenotype is rescued by co-infection with wild-type (WT) parasites *in vivo*, independent of host functional adaptive immunity or key pro-inflammatory cytokines. These data show that the major function of MYR1-dependent secreted proteins is not to protect the parasite from clearance within infected cells. Instead, MYR-dependent proteins generate a permissive niche in a paracrine manner, which rescues ΔMYR1 parasites within a pool of CRISPR mutants in mice. Our results highlight an important limitation of otherwise powerful *in vivo* CRISPR screens and point towards key functions for MYR1-dependent *Toxoplasma*-host interactions beyond the infected cell.

## Introduction

Pooled CRISPR screens have been an extraordinarily powerful genetic tool to identify gene function in an unbiased manner using negative or positive selection. They have been applied in various cell culture conditions and *in vivo*, in combination with different genetic or chemical bottlenecks to identify genes in a specific setting (*Bock et al., 2022*). Fitness-conferring genes are identified by assessing the relative abundance of cells with different genetic perturbations within a pool of mutants upon selective pressure.

We have previously performed pooled CRISPR screens in mice with the intracellular parasite *Toxoplasma gondii*, to identify exported parasite proteins that are important for survival in his natural host (*Young et al., 2019*; *Butterworth et al., 2022*). For the majority of tested genes, the phenotypes observed in pooled CRISPR screens are concordant to those observed in single-strain infections using clonal mutants for those genes. To our surprise, however, we found that while some *Toxoplasma* genes are required for survival in single-strain mutant KO murine infections, this fitness defect phenotype is lost when the same mutants are part of a heterogenous mutant pool used for CRISPR screens in mice. We hypothesised that individual mutants can be rescued by paracrine effects triggered by other parasites in the pool, pointing towards a potential limitation of pooled CRISPR screens to detect a subset of important virulence factors in *Toxoplasma*. This limitation likely is not restricted to *Toxoplasma*, or other infectious contexts, but CRISPR screens in general where paracrine effects are operating.

*Toxoplasma* is a ubiquitous parasitic pathogen in all warm-blooded animals. *Toxoplasma* infection is widespread in livestock and wild animals, as well as in humans, where one third of the population is estimated to be seropositive (*Pappas et al., 2009*). Following oral infection by oocysts or latent stage tissue cysts, the parasite grows as tachyzoites in intermediate hosts, before disseminating to distal organs. The host immune response is pivotal to reduce parasite burden caused by rapidly proliferating tachyzoites. However, it is not sufficient to completely clear infection, as surviving parasites can differentiate into the chronic bradyzoite forms and establish tissue cysts in the central nervous system and skeletal muscle cells (*Dubey et al., 1998*). During the onset of infection, recognition of *Toxoplasma*-derived pathogen-associated molecular patterns (PAMPs) by host innate immune cells triggers the production of IL-12, amongst other pro-inflammatory mediators (*Sasai and Yamamoto, 2019*). IL-12 has a protective role during toxoplasmosis, primarily by triggering IFN-γ production in lymphocytes and thereby linking innate and adaptive immunity during infection (*Gazzinelli et al., 1994*). IFN-γ is a pivotal cytokine in conferring resistance against *Toxoplasma* infection since it induces expression of interferon-stimulated genes (ISGs) that limit intracellular parasite proliferation and curtail infection (*Saeij and Frickel, 2017*; *MacMicking, 2012*; *Hunn et al., 2011*). These antimicrobial effects are enhanced by other pro-inflammatory cytokines. An example is TNF, which acts as a co-stimulatory signal to trigger IFN-γ production by NK cells exposed to the parasite, and boosts the antimicrobial activity of IFN-γ-activated macrophages (*Sibley et al., 1991*; *Langermans et al., 1992*; *Sher et al., 1993*). Following *Toxoplasma* infection, host cells can also secrete chemokines such as CCL2, which drives recruitment of CCR2+Ly6C^high inflammatory monocytes from the bone marrow to the infected sites (*Robben et al., 2005*; *Dunay et al., 2008*). There, they can differentiate into monocyte-derived dendritic cells or macrophages and act as an extra line of defence against the parasite (*Ruiz-Rosado et al., 2016*; *Goldszmid et al., 2012*).

In order to survive clearance by host immune cells and to disseminate within the infected host, *Toxoplasma* relies on an array of over 250 secreted proteins (*Barylyuk et al., 2020*). Following invasion, *Toxoplasma* replicates inside a parasitophorous vacuole (PV) separated from the host cytoplasm by the PV membrane (PVM). During and after host cell invasion, *Toxoplasma* secretes proteins from the rhoptries and the dense granules. Secreted proteins from these organelles are not only pivotal for vacuole establishment, but also nutrient uptake, reprogramming of the infected cell and protection against the host immune response (*Hakimi et al., 2017*). In order to exert their effect, some *Toxoplasma* proteins secreted from dense granules must cross the PVM, likely via a multi-protein translocon that depends on MYR1 (*Franco et al., 2016*). GRA16, a dense granule effector that relies on MYR-dependent export to exit the PV and reach the host cytosol, was shown to drive upregulation of the transcription factor c-Myc in host cells (*Panas and Boothroyd, 2020*). After the identification of MYR1 other putative components of the MYR translocon – MYR2, MYR3, MYR4, ROP17 – were identified, using their ability to trigger GRA16-dependent c-Myc induction in infected cells as a surrogate (*Marino et al., 2018*; *Panas et al., 2019a*; *Cygan et al., 2020*). It was subsequently suggested

that most, if not all exported dense granule proteins that reach the host cell cytosol might depend on functional MYR1 for translocation (*Panas et al., 2019a*; *Sangaré et al., 2019*). To date, seven virulence factors have been shown to be MYR1-dependent: IST, NSM, HCE1/TEEGR, GRA16, GRA18, GRA24, and GRA28. Once exported, these parasite proteins can interfere with host cell transcription and contribute to the establishment of permissive niches for *Toxoplasma* proliferation and survival via multiple downstream mechanisms. Examples include boosting resistance against IFN-γ-dependent antimicrobial mechanisms (IST *Olias et al., 2016*; *Gay et al., 2016*), arresting the host cell cycle (HCE1/TEEGR *Panas et al., 2019b*; *Braun et al., 2019*), inhibiting programmed host cell death pathways (NSM *Rosenberg and Sibley, 2021*) and modulating cytokine/chemokine secretion (GRA16, GRA18, GRA24, GRA28 *Bougdour et al., 2013*; *He et al., 2018*; *Braun et al., 2013*; *Ten Hoeve et al., 2022*).

Given the combined importance of exported dense granule effectors, it is not surprising that the vast majority of transcriptional changes in *Toxoplasma*-infected human fibroblasts are dependent on MYR1 (*Naor et al., 2018*), and that isogenic MYR1-deficient strains are avirulent in mice (*Franco et al., 2016*). In pooled *in vivo* CRISPR screens, however, we have shown that parasites lacking MYR1 had no fitness defect in a five-day infection experiment within the mouse peritoneum (*Young et al., 2019*; *Butterworth et al., 2022*). This is in agreement with results from *in vivo* CRISPR screens performed in other labs using different parasite strains and mouse backgrounds where no (*Sangaré et al., 2019*; *Tachibana et al., 2023*), or only very mild growth defects (*Giuliano et al., 2024*) have been observed for MYR1. This divergence of results between isogenic infection, which clearly shows an important role for MYR1 in murine infections, and the pooled CRISPR screens, which show no defect of MYR1 mutants, led us to hypothesise that: (1) MYR1, and therefore proteins that rely on the MYR complex for translocation, are not essential for the cell-autonomous survival of parasites in macrophages, the main cell type infected in the first days of a peritoneal infection (*Jensen et al., 2011*); and (2) ΔMYR1 parasites within a pool of mutants may be rescued by MYR1-competent parasites via a paracrine effect, setting up a parasite-permissive immune environment in which MYR1-deficient mutants can thrive.

Here, we verify both hypotheses, showing that MYR1 is not important for the *Toxoplasma* cell-autonomous survival within macrophages, and deploy co-infection strategies proving that MYR1-competent parasites can trans-rescue the growth defect of ΔMYR1 parasites *in vivo*. This rescue does not depend on the host adaptive immune response and surprisingly still occurs despite high levels of key pro-inflammatory cytokines. This knowledge is paramount to understand the biological function of MYR1-driven rewiring of the host cell, and consequently the function of MYR1-dependent effector proteins. It also highlights an important limitation of otherwise powerful *in vivo* pooled CRISPR screens in *Toxoplasma*, where loss-of-function of a protein in one parasite can be rescued by other parasites in the pool. This limitation likely extends to CRISPR screens in other biological contexts in which paracrine effects operate.

## Results

### MYR1 is not essential for survival within IFN-γ-stimulated macrophages

To test whether MYR1 was required for the parasite cell-autonomous survival in immune cells, we infected IFN-γ-primed bone marrow-derived macrophages (BMDMs) with WT or MYR1 KO parasites established in the type II Prugniaud (Pru) genetic background, and quantified infected cells after 24 h. As GRA12 was shown to be required for parasite survival in IFN-γ-stimulated macrophages (*Fox et al., 2019*) and had a strong growth defect in our pooled *in vivo* CRISPR screen (*Young et al., 2019*), we included a ΔGRA12 strain in these experiments as positive control. As negative control, we included parasites lacking IST (*Figure 1—figure supplement 1*), a known MYR1-dependent factor that inhibits the induction of interferon-stimulated genes. IST protects the parasites against intracellular restriction when the infection precedes IFN-γ activation, but not if *Toxoplasma* infects primed cells (*Olias et al., 2016*; *Gay et al., 2016*). As expected, ΔGRA12 parasites showed a significant reduction in the proportion of infected cells in the presence of IFN-γ (*Figure 1a*). Infection with ΔMYR1 was comparable to ΔIST and not significantly different to that of WT parasites, although a slight increase in restriction was observed for ΔIST and ΔMYR1. Our results are in line with findings from Wang and colleagues where

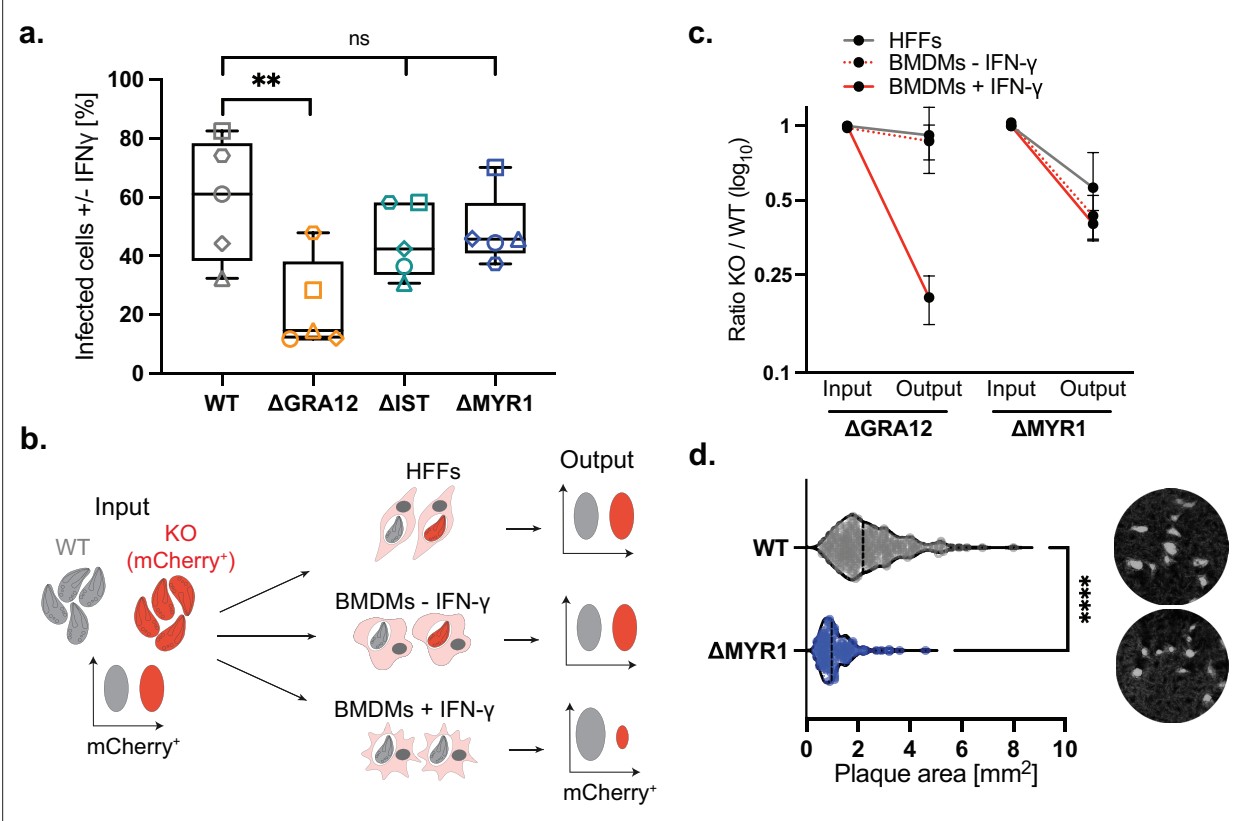

**Figure 1.** MYR1 and MYR1-dependent factors are not contributing to *in vitro Toxoplasma* survival in IFN-γ-activated macrophages. (**a**) BMDMs were stimulated with 100 U/ml IFN-γ for 24 hr or left untreated, before infection with mCherry-expressing *Toxoplasma* strains for 24 hr. Cells were fixed and imaged by high-content imaging, and the percentage of infected cells in IFN-γ-treated BMDMs compared to untreated controls is shown. WT refers to PruΔUPRT. The box-plot shows the median value ± SD and the whiskers show minimum and maximum values. Significance was tested with the One-way Anova test with the Benjamini, Krieger and Yekutieli FDR correction, n=5. (**b**) Schematic of the flow cytometer-based growth competition assay of an IFN-γ-dependent effector (e.g. GRA12). HFFs and BMDMs were co-infected with equal amounts of colourless WT and mCherry-expressing ΔGRA12 or ΔMYR1 strains. BMDMs were stimulated with 100 U/ml IFN-γ for 24 hr or left untreated before infection. The mCherry signal was quantified by flow cytometry analysis and the ratio between the strains after two passages (output) was compared to the input. (**c**) Growth of competing mutant and WT parasites was assessed by flow cytometry. The normalised ratios of output versus input are shown. The average of two independent experiments with technical triplicates is shown. Bars display mean ± SD. (**d**) Violin plot of the plaque size of parental PruΔKU80 (n=202) and derived PruΔMYR1 (n=95) strains assessed in two independent experiments. The bar represents the median value and representative images are reported on the right. Significance was tested using a two-tailed unpaired Welch's t-test, ** p<0.01, **** p<0.0001.

The online version of this article includes the following source data and figure supplement(s) for figure 1:

**Source data 1.** Raw data of the high-content imaging quantification of *Toxoplasma*-infected BMDMs pre-treated or not with IFN-γ displayed in *Figure 1a*.

**Source data 2.** Raw data of the flow cytometry-based competition assay experiments in BMDMs pre-treated or not with IFN-γ, and human fibroblasts displayed in *Figure 1c*.

**Source data 3.** Combined plaque sizes data of PruΔMYR1 parasites compared to the parental strain displayed in *Figure 1d*.

**Source data 4.** PDF file containing the original plaque images analysed for *Figure 1d*.

**Source data 5.** Original cropped plaque images displayed in *Figure 1d*.

**Figure supplement 1.** Validation of the created PruΔIST strain.

**Figure supplement 1—source data 1.** PDF file containing the original agarose gel image displayed in *Figure 1—figure supplement 1*.

**Figure supplement 1—source data 2.** Original agarose gel image displayed in *Figure 1—figure supplement 1*.

deletion of MYR1 in a more virulent type I strain similarly causes no fitness defect in IFN-γ-activated macrophages (*Wang et al., 2020*).

To further validate the role of virulence factors in conferring parasite resistance against IFN-γ-mediated host defence, we assessed replication of ΔMYR1 or ΔGRA12 in competition with WT parasites

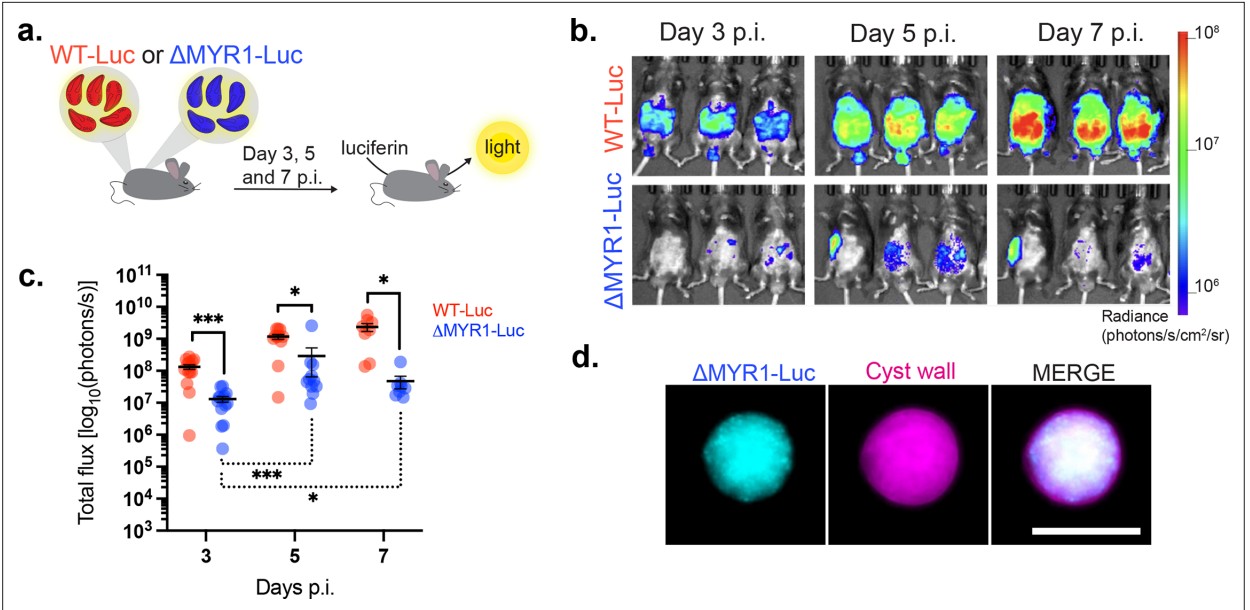

**Figure 2.** ΔMYR1 parasites expand in the murine host and form cysts *in vivo*. (**a**) Mice were infected i.p. with luciferase-expressing *Toxoplasma* strains, WT-Luc or ΔMYR1-Luc, and whole-body intravital imaging was performed at days 3, 5, and 7 p.i. (**b**) Representative whole-body intravital imaging. (**c**) Graph shows the total bioluminescence signal converted to a logarithmic scale from two independent experiments. Statistical differences between the two strains at each timepoint were tested with the Mixed-effects model (REML) with Sidak post-hoc tests (continued lines). Significance for the ΔMYR1-Luc growth over time was tested with a One-Way Anova test (dotted line). Bars display mean ± SEM. Number of mice per group: day 3 p.i.: n=14; day 5 p.i.: n=11; day 7 p.i.: n=8. * p<0.05, *** p<0.001. (**d**) Representative fluorescent image of a brain cyst from a mouse infected with ΔMYR1-Luc. mCherry-expressing ΔMYR1-Luc parasites were detected by microscopy. Cyst wall was stained with the FITC-conjugated lectin DBA. The scale bar represents 25 µm.

The online version of this article includes the following source data and figure supplement(s) for figure 2:

**Source data 1.** PDF containing the intravital images displayed in *Figure 2b*.

**Source data 2.** Original images displayed in *Figure 2b*.

**Source data 3.** Raw data of the total bioluminescent signal from intravital imaging displayed in *Figure 2c*.

**Source data 4.** PDF containing the original image of the *Toxoplasma* cyst displayed in *Figure 2d*.

**Source data 5.** Original image of the *Toxoplasma* cyst displayed in *Figure 2d*.

**Figure supplement 1.** Validation of the created Luciferase-expressing strains.

**Figure supplement 1—source data 1.** PDF file containing the original agarose gel image displayed in *Figure 2—figure supplement 1*.

**Figure supplement 1—source data 2.** Original agarose gel image displayed in *Figure 2—figure supplement 1*.

over two lytic cycles in IFN-γ-treated or unprimed BMDMs, and in HFFs as control (*Figure 1b*). As expected, ΔGRA12 was outcompeted by WT parasites exclusively in BMDMs pre-stimulated with IFN-γ, confirming its role to survive the cytokine-mediated clearance in macrophages (*Figure 1c*). On the contrary, ΔMYR1 parasites displayed a fitness defect compared to WT parasites, regardless of the infected cell type and independently of IFN-γ treatment (*Figure 1c*). This result is in line with the smaller size of the ΔMYR1 plaques established in HFFs compared to the parental strain (*Figure 1d*), which recapitulates published data (*Marino et al., 2018*).

## MYR1 mutants expand during the course of infection and can form tissue cysts *in vivo*

To investigate the growth of ΔMYR1 parasites *in vivo* and follow infection over time, we generated parasite strains expressing firefly luciferase (Luc) in the wild-type (WT-Luc) or ΔMYR1 backgrounds (ΔMYR1-Luc, *Figure 2—figure supplement 1*). We injected mice with 25,000 tachyzoites and monitored parasite growth over 7 days by intravital imaging (*Figure 2a*). As expected from the reduced *in vitro* growth phenotype in HFFs and BMDMs, ΔMYR1-Luc showed reduced bioluminescent signal compared to WT-Luc parasites (*Figure 2b and c*), confirming that the growth defect *in vitro* results

in analogous phenotypes *in vivo*. Nevertheless, MYR1-deficient parasites were still able to expand in mice, as bioluminescence signal increased from day 3 to day 5 post infection (*Figure 2b and c*). This initial increase of growth is similar to what was observed for ΔIST parasites (*Olias et al., 2016*; *Gay et al., 2016*). ΔMYR1-Luc-infected mice that survived the acute phase of infection for 4 weeks p.i. carried a low number of cysts in the brain (3.4 cysts/brain on average; 0–13 cysts/brain detected, *Figure 2d*). This is similar to previous observations (*Young et al., 2019*) indicating that expression of luciferase does not decrease the ability to produce cysts and further supports the notion that ΔMYR1 parasites are able to form cysts, albeit at low numbers. As mice infected with WT parasites succumbed during the acute stage of infection, we cannot compare cyst numbers between the two strains.

## MYR1-dependent secreted factor(s) rescue ΔMYR1 *in vivo* growth defect via a paracrine mechanism, independent of host adaptive immunity

We have shown that ΔMYR1 parasites have reduced *in vitro* and *in vivo* growth, despite not having any significant growth defect when in a pool with other mutants. Therefore, we hypothesised that MYR1-competent parasites within the KO pool generate a permissive environment that ultimately promotes growth of the ΔMYR1 mutants in a paracrine fashion. To test this hypothesis, we performed co-infection experiments where mice were injected with an inoculum containing a 20:80 ratio of ΔMYR1-Luc parasites with either WT parasites or ΔMYR1 mutants not expressing luciferase (*Figure 3a*). This setting allows us to assess if the presence of WT parasites affects growth of ΔMYR1-Luc parasites within the peritoneum. Our results show that ΔMYR1-Luc parasites proliferate better in mice co-infected with WT parasites than with ΔMYR1 parasites (*Figure 3b and c*). These results support a paracrine role of MYR1-mediated effectors *in vivo* (*Figure 4*).

To understand what mediates this trans-rescue phenotype, we assessed the production of selected pro-inflammatory cytokines important during *Toxoplasma* infection. Mice infected with a mix of ΔMYR1:WT or ΔMYR1:ΔMYR1 parasites display comparable levels of IL-12p40 in both peritoneum and serum at day 7 p.i. However, ΔMYR1:WT infections elicited higher levels of CCL2/MCP-1, TNF and especially IFN-γ compared to the ΔMYR1:ΔMYR1 mix (*Figure 3d*, *Figure 3—figure supplement 1*).

Considering that ΔMYR1-Luc growth is rescued even when high levels of IFN-γ are produced in the peritoneal cavity, we wanted to assess whether disrupting IFN-γ-producing cell populations would impact the trans-rescue phenotype. Lymphocytes, in particular CD8+, Th1-committed CD4+ and γδ T cells, are the main sources of IFN-γ during *Toxoplasma* infection (*Nishiyama et al., 2020*). Thus, we applied the same mixed infection strategy in Rag2-deficient mice that do not produce mature T and B cells, and therefore fail to deploy adaptive immune responses (*Shinkai et al., 1992*; *Figure 3e*). As expected, an overall higher parasitaemia was detected in *Rag*2−/− mice when compared to WT mice, due to the known role of the T and B cells to control infection. Nevertheless, in both *Rag*2−/− and WT control mice ΔMYR1-Luc parasites proliferate more when mixed with WT parasites than with ΔMYR1 parasites (*Figure 3f*). These results confirm that the host adaptive response is not essential for the trans-rescue of MYR1-deficient *Toxoplasma* during acute infection.

## Discussion

In this work, we show that deletion of MYR1, and by extension MYR1-dependent effectors, does not impact the ability of *Toxoplasma* to initiate an infection in mice and survive in IFN-γ-stimulated murine macrophages. While clonal ΔMYR1 mutants have a significantly reduced growth compared to WT parasites *in vivo*, co-infection with WT parasites increased their ability to proliferate. This finding supports results from pooled CRISPR-Cas9 screens, where loss-of-function mutants of *Myr1* and other genes previously shown to be involved in PV translocation of effector proteins, show no (*Young et al., 2019*; *Butterworth et al., 2022*; *Sangaré et al., 2019*; *Tachibana et al., 2023*) or only relatively minor fitness defects in mice (*Giuliano et al., 2024*). The rescue phenotype observed in mixed infections is very unlikely to occur through co-infection of host cells by WT and ΔMYR1 parasites, as in the peritoneal exudate from infected mice less than 2% of all infected cells were co-infected (data not shown). As such, our data indicates that some MYR1-dependent effectors cause changes in infected murine cells, that in turn provide a favourable environment for parasite growth in a paracrine manner

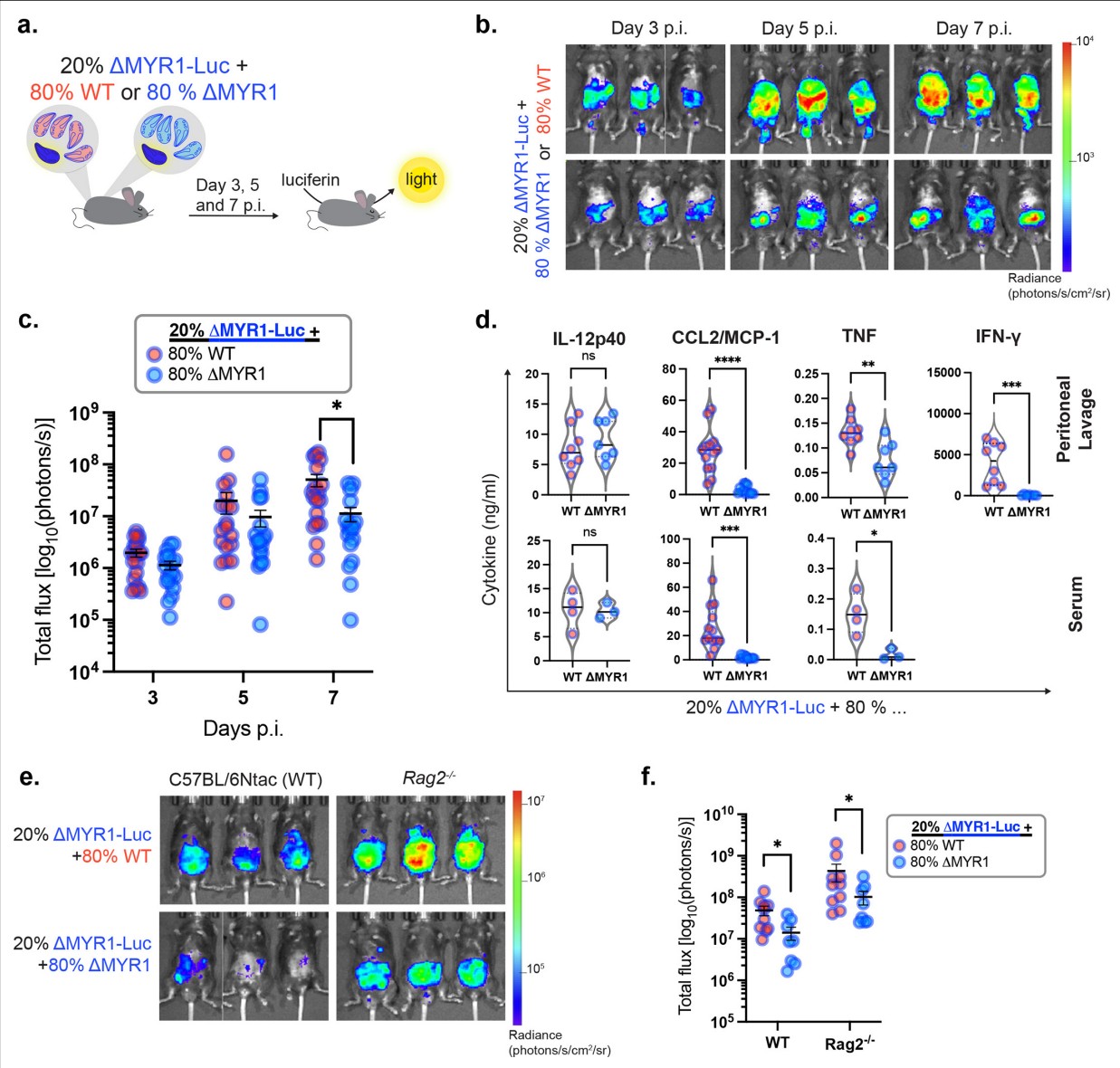

**Figure 3.** The *in vivo* growth defect of ΔMYR1 is rescued by the presence of MYR1-competent parasites, independently of functional host adaptive immunity. (**a**) Mice were infected i.p. with a mixed inoculum of *Toxoplasma* tachyzoites, containing a 20:80 ratio of luciferase-expressing ΔMYR1-Luc to WT or ΔMYR1 strains that do not express luciferase. Growth of ΔMYR1-Luc was monitored at 3, 5, and 7 days p.i. by whole-body intravital imaging. (**b**) Representative whole-body intravital imaging. (**c**) Total bioluminescence signal converted to a logarithmic scale from mice co-infected with ΔMYR1-Luc:WT (n=18) or ΔMYR1-Luc:ΔMYR1 (n=17). Graph shows cumulative data from four independent experiments. Bars display mean ± SEM. Significance was tested with a Two-way Repeated Measures ANOVA with Sidak post-hoc tests. (**d**) IL-12p40, CCL2/MCP-1, TNF and IFN-γ levels were detected in peritoneal lavage and serum of animals infected with mixed *Toxoplasma* inoculum (20:80 ratio) at day 7 p.i. by ELISA. The violin-plots show the median (continued black line) and quartiles (dotted grey lines). Significance was tested with the Welch's t-test (TNF, CCL2/MCP-1 and IL-12p40 ELISAs) and Mann-Whitney test (IFN-γ ELISAs). Samples from two to three independent experiments were assayed. Number of samples for CCL2: WT n=13 and ΔMYR1 n=12. Number of samples for other cytokines: peritoneal lavage samples: WT n=8 and ΔMYR1 n=7; serum samples: WT n=4 and ΔMYR1 n=3. Data with higher differences between inocula (CCL2 and IFN-γ) are displayed in a logarithmic scale in *Figure 3—figure supplement 1*. (**e**) Representative whole-body intravital imaging at day 7 p.i. of C57BL/6Ntac (WT) or RAG2-deficient (*Rag2-/-*) mice infected i.p. with a mixed inoculum containing a 20:80 ratio of luciferase-expressing ΔMYR1-Luc tachyzoites to WT or ΔMYR1 strains that do not express luciferase. (**f**) Total bioluminescence signal converted to a logarithmic scale from mice co-infected with ΔMYR1-Luc:WT (n=10) or ΔMYR1-Luc:ΔMYR1 (n=8). Graph shows cumulative data from two independent experiments. Bars display mean ± SEM. Significance was tested with the Multiple Mann-Whitney test with a False Discovery Rate approach by a Two-stage step-method. * p<0.05, ** p<0.01, *** p<0.001, **** p<0.0001.

The online version of this article includes the following source data and figure supplement(s) for figure 3:

**Source data 1.** PDF containing the intravital images displayed in *Figure 3b*.

*Figure 3 continued on next page*

*Figure 3 continued*

**Source data 2.** Original images displayed in *Figure 3b*.

**Source data 3.** Raw data of the total bioluminescent signal from intravital imaging displayed in *Figure 3c*.

**Source data 4.** Raw data of the ELISA results displayed in *Figure 3d*.

**Source data 5.** PDF containing the intravital images displayed in *Figure 3e*.

**Source data 6.** Original images displayed in *Figure 3e*.

**Source data 7.** Raw data of the total bioluminescent signal from intravital imaging displayed in *Figure 3f*.

**Figure supplement 1.** Infection with ΔMYR1 parasites elicits less IFN-γ and CCL2 compared to wild-type paraites.

(*Figure 4*). This is important as: (1) it shows that the major transcriptional changes caused by MYR-dependent effectors are not required for *Toxoplasma* to survive cell-autonomous immune responses in IFN-γ-primed cells; (2) Paracrine rescue in pooled CRISPR-Cas9 screens may mask a significant amount of proteins required for *Toxoplasma* survival; (3) It is likely that this 'paracrine masking effect' can be found in CRISPR screens in other biological contexts, for example: other host-microbe inter-action screens, and possibly even in non-infectious biological contexts, such as pooled CRISPR screen approaches to study cancer immunity and to discover regulators of innate and adaptive immunity crosstalk (reviewed in *Shi et al., 2023*; *Holcomb et al., 2022*).

We found that the presence of WT parasites in a mixed inoculum with ΔMYR1 mutants elicited significantly higher levels of IFN-γ and TNF in the peritoneum than ΔMYR1 mutants alone. As IFN-γ was shown to be a major cytokine to limit *Toxoplasma* growth in a cell-autonomous manner, and that TNF acts mainly by enhancing the antimicrobial effects in IFN-γ-activated cells, one would expect that ΔMYR1-Luc parasites would be more restricted when injected in combination with WT parasites. However, we observed the opposite, as higher ΔMYR1-Luc parasitaemia was observed in ΔMYR1:WT than in ΔMYR1:ΔMYR1 mixes. These data provide further support that MYR1, and by extension MYR1-dependent effectors, do not protect *Toxoplasma* from the IFN-γ-mediated intracellular clearance in mice.

What could be the driving force of the rescue? We show that the adaptive immune system plays no role, pointing towards cells of the innate immune system. Higher CCL2 levels in mice infected with ΔMYR1:WT suggests higher inflammatory Ly6C[high] monocytes recruitment to infected tissues (*Robben et al., 2005*; *Dunay et al., 2008*). These cells could be responsible for the high levels of TNF and IFN-γ detected. While monocytes have been shown in multiple reports to be important for limiting *Toxoplasma* upon exposure to IFN-γ in infected niches (*Robben et al., 2005*; *Goldszmid et al., 2012*; *Mordue and Sibley, 2003*), it is possible that recruited monocytes also provide an important reservoir

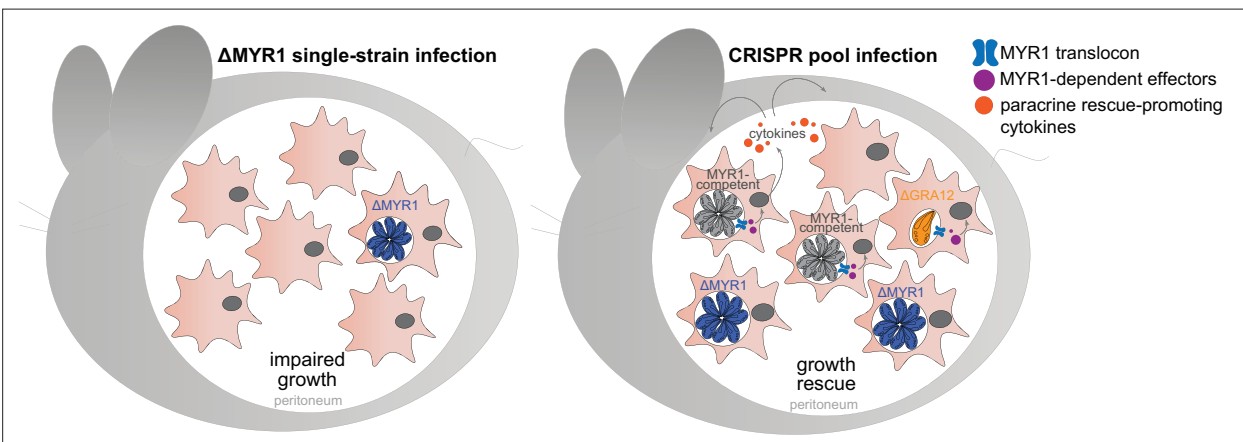

**Figure 4.** Schematic of the difference between single-strain *Toxoplasma* ΔMYR1 infections and CRISPR pool infections *in vivo*. ΔMYR1 parasites do not expand within the peritoneum and are rapidly cleared (left panel). ΔMYR1 parasites within a CRISPR pool are rescued by MYR-competent parasites in a paracrine manner (right panel). MYR-dependent factors cross the MYR-translocon and affect growth within the peritoneal environment in trans, through cytokine secretion and recruitment of host cells. However, *Toxoplasma* virulence factors that act at the cell-autonomous level, such as GRA12, cannot be rescued in trans.

for parasites to grow in the peritoneum. However, how ΔMYR1 parasites are eventually cleared in homogenous infections in the absence of high levels of CCL2 is not yet known.

Individual MYR-related effectors that may be responsible for the paracrine rescue have not been investigated here and we hypothesise that the phenotype is likely the concerted result of multiple effectors that affect cytokine secretion. For example, previous studies showed that both GRA18 and GRA28 can induce release of CCL22 from infected cells (*He et al., 2018*; *Rudzki et al., 2021*), while GRA16 and HCE1/TEEGR impair NF-kB signalling and the potential release of pro-inflammatory cytokines such as IL-6, IL-1β, and TNF (*Braun et al., 2019*; *Seo et al., 2020*). Regardless of the effector(s), our results highlight an important novel function of MYR1-dependent effectors by establishing a supportive environment in trans for *Toxoplasma* growth within the peritoneum.

We further confirm previous results using luciferase-expressing ΔMYR1 parasites that MYR1 appears to be dispensable for the formation and persistence of latent *Toxoplasma* stages per se. As MYR1 has been demonstrated to be dispensable for stage conversion *in vitro* (*Seizova et al., 2022*), the relatively low number of cysts is likely explained by a failure of ΔMYR1 parasites to efficiently disseminate and/or persist within the murine host (*Ten Hoeve et al., 2022*). In alternative, we hypothesise that the absence of CCL2 in ΔMYR1 infections limits recruitment of host cells that *Toxoplasma* can potentially use as vehicles to reach the brain.

This work also highlights the limitations of restriction-based CRISPR-screens in capturing the variety of pathogen's mechanisms to survive host clearance. Novel applications of the CRISPR screens, for example in combination with single-cell RNA sequencing (*Butterworth et al., 2023*) or with functional assays to explore immunological contexts (*Shi et al., 2023*) could help understand how infected cells affect the neighbouring environment to support infection, and contribute to parasite survival, dissemination and persistence within the host. Here, we show that MYR1-dependent proteins play a critical role in promoting a favourable environment for growth beyond the infected cell in a paracrine manner. This is different to the injection of rhoptry effector proteins by *Toxoplasma* into cells it does not invade, which requires parasite-host cell contact (*Koshy et al., 2012*), and provides a novel angle on how the parasite can systematically alter the host environment in its favour during infection.

Our work draws attention to an understudied aspect of pathogen manipulation in complex multicellular settings which warrants further studies. The limitation we highlight here, that mutant phenotypes can be masked in pooled CRISPR screens, likely extends to other experimental setups where paracrine effects are possible.

## Materials and methods
### Cell culture and parasite strains
Primary human foreskin fibroblasts (HFFs; ATCC SCRC-1041) were maintained in Dulbecco's modified Eagle's Medium (DMEM) with 4.5 g/L glucose and GlutaMAX-1 (Gibco Thermo Fisher Scientific) supplemented with 10% foetal bovine serum (FBS; Gibco Thermo Fisher Scientific) at 37 °C and 5% $CO_2$. To generate bone marrow-derived macrophages (BMDMs), bone marrow cells were extracted from femurs of C57BL/6 J mice and differentiated to macrophages for 7 days at 37 °C and 5% $CO_2$. Briefly, bone marrow cells were seeded in 15 $cm^2$ Petri dishes (Corning) and differentiated into BMDMs in RPMI 1640 medium (Gibco Thermo Fisher Scientific) supplemented with 20% L929 conditioned media (containing the murine macrophage colony-stimulating factor [M-CSF]), 50 μM 2-mercaptoethanol, Penicillin/Streptomycin (Thermo Fisher Scientific), and 10% FBS. For experiments BMDMs were grown in the above RPMI medium but lacking 2-mercaptoethanol (called working media further on). *Toxoplasma gondii* strains PruΔKU80 (*Fox et al., 2011*), PruΔUPRT::mCherry (*Butterworth et al., 2022*), PruΔGRA12::mCherry (*Young et al., 2019*), PruΔMYR1::mCherry (*Young et al., 2019*) and derived strains were maintained in confluent HFFs and passaged by syringe lysis through 23 G needles every 2–3 days.

### Generation of parasite lines
$10^6$ freshly lysed parasites were transfected with 20–25 μg of DNA by electroporation using the 4D-Nucleofector (Lonza) with previously optimised protocols as described in *Young et al., 2019*. All primers used are listed in *Supplementary file 1*. To generate pSAG1::Cas9sgIST the IST gRNA sequence was inserted into pSAG1::Cas9sgUPRT (*Shen et al., 2014*) by inverse PCR using primers 1&2. To generate

the IST KO (PruΔIST::mCherry), PruΔKU80 parasites were co-transfected with pSAG1::CAS9sgIST and a Pro-GRA1::mCherry::T2A::HXGPRT::Ter-GRA2 construct amplified with primers 3&4 containing 40 bp homology regions to the 5′- and 3′- untranslated regions of IST (ToxoDB TGME49_240060). 24 hr after transfection, 50 μg/ml Mycophenolic acid (Merck) and Xanthine (Sigma) (M/X) were added to select for integration, and a clonal culture was verified by PCR with primers 5–8. To generate Luciferase expressing parasite lines, PruΔKU80 or PruΔMYR1::mCherry were co-transfected with pSAG1::CAS9sgUPRT and PciI digested pUPRT-ffLucHA to insert a HA-tagged Firefly luciferase gene (LucHA) into the *Uprt* locus and establish WT-Luc and ΔMYR1-Luc strains respectively. To generate pUPRT-ffLucHA, the GRA1 promoter region and firefly Luciferase sequences were amplified from pGRA (*Coppens et al., 2006*) and pDHFR-Luc (*Saeij et al., 2005*) plasmids respectively using primers 9&10 and 11&12 and combined with BamHI/Xma digested pUPRT-HA in a Gibson reaction. 24 hr post transfection 20 μg/ml FUDR (Sigma) was added to select for disruption of the *Uprt* locus and clones verified by PCR with primers 13&14.

## Parasite growth in BMDMs

$5x10^4$ BMDMs were seeded per well in 96-well Ibitreat black μ-plates (Ibidi GmbH) in working media. The following day cells were either treated with IFN-γ (100 U/mL, Peprotech) or left untreated. 24 hr later cells were infected with the strains in triplicate at a multiplicity of infection (MOI) of 0.3, and the plate was centrifuged at 210 × *g* for 3 min. At 3 hr post infection (p.i.), the medium was replaced to remove non-invaded parasites. Cells were fixed in 4% paraformaldehyde (PFA) at 24 hr p.i., washed in PBS and stained with Far Red Cell Mask (1:2,000, Thermofisher Scientific). Plates were imaged on an Opera Phenix High-Content Screening System (PerkinElmer) with a 40 x NA1.1 water immersion objective. 38 fields of view with 10 planes were imaged per well. Analysis was performed on a maximum projection of the planes and the percentage of infected cells was quantified over triplicate samples similarly to previously established protocols (*Butterworth et al., 2022*; *Lockyer et al., 2023*).

## Competition assay

$5x10^5$ BMDMs were seeded in 12-well plates in working media. The following day cells were either treated with 100 U/ml IFN-γ or left untreated. 24 hr post treatment BMDMs were infected with a 1:1 mix of either WT (PruΔKU80) and PruΔGRA12::mCherry parasites or WT and PruΔMYR1::mCherry parasites at a MOI of 0.3 ($1.5x10^5$ parasites per well). Confluent HFFs in 12-well plates were similarly infected to evaluate defects in growth of the KO parasites. A sample of input parasites were fixed in 4% PFA to verify the starting ratio. 48 hr p.i. cells were scraped and passed through a 27 G needle, and the parasites inoculated onto HFF monolayers and grown for a further 3–4 days. For flow cytometry analysis, cells were lysed with 23 G needles and the parasites passed through a 5 μm filter before fixation in 4% PFA and staining with Hoechst 33342 (Thermo Fisher Scientific). Samples were analysed on a BD LSR Fortessa flow cytometer and with FlowJo software v10. Hoechst 33342 was excited by a 355 nm laser and detected by a 450/50 band pass filter. mCherry was excited by a 561 nm laser and detected by a 600 long pass filter and a 610/20 band pass filter. To eliminate debris from the analysis, events were gated on forward scatter, side scatter and Hoechst 33342 fluorescence. Parasites were identified by their nuclear staining and KO were discriminated by their mCherry signal. The ratios of KO/WT parasites (mCherry$^+$/ mCherry$^-$) from two independent experiments, each in technical triplicates for each condition, were calculated and normalised by dividing by the input ratio, to allow comparison between strains and biological replicates.

## Plaque assay

HFF were grown to confluency in T25 flasks and infected with 200 parasites to grow undisturbed for 10 days. Cells were fixed and stained in a solution with 0.5% (w/v) crystal violet (Sigma), 0.9% (w/v) ammonium oxalate (Sigma), 20% (v/v) methanol in distilled water, then washed with tap water. Plaques were imaged on a ChemiDoc imaging system (BioRad) and measured in FIJI (*Schindelin et al., 2012*).

### *In vivo* infections

Male and female mice, aged 6–12 weeks were used in this study. For experiments, animals were sex- and age-matched. Mice were injected intraperitoneally (i.p.) with a total of 25,000 *Toxoplasma gondii* tachyzoites in 200 µL PBS either as a single-strain inoculum (100% PruΔUPRT::LucHA or PruΔMYR1::LucHA), or as a mixed strain inoculum including 5,000 tachyzoites of luciferase-expressing strain PruΔMYR1::LucHA::mCherry (20%) and 20,000 tachyzoites of PruΔKU80 or PruΔMYR1::mCherry strains (80%). Mice were monitored and weighed regularly throughout the experiments. Parasite *in vivo* growth was monitored by intravital imaging (IVIS) at days 3, 5, and 7 p.i. Mice were injected i.p. with 100 µL of 30 mg/ml luciferin (PerkinElmer) in PBS and were anaesthetised (isoflurane 5% for induction and 2.5% afterwards) 15 min prior to bioluminescent imaging on an IVIS Spectrum CT (Perkin-Elmer). Animals were euthanised at day 7 p.i. or at humane endpoints, blood was collected by cardiac puncture and peritoneal exudate cells were harvested by peritoneal lavage through injection of 1 mL PBS i.p. Blood was transferred to serum separating tubes and centrifuged at 10,000 rpm for 10 min at 4 °C to isolate serum, and peritoneal exudate was spun at 500 *g* for 5 min to remove the cellular component. IL-12p40 (#88-7120-88), TNF (#88-7324-88), IFN-γ (#88-7314-88) and CCL2/MCP-1 (#88-7391-88) levels were detected on serum and/or in suspension at peritoneal exudates of infected mice by ELISA, following manufacturer's protocol (Thermo Fisher). To confirm cyst formation, the brain of mice surviving infection with PruΔMYR1::LucHA::mCherry tachyzoites were homogenised in 1 mL PBS and 300 µL of the sample were stained with FITC-conjugated Dolichos Biflorus Agglutinin (DBA; 1: 200; Vector Laboratories #RL-1031) for 1 hr at room temperature. Fluorescently labelled cysts were counted using a Ti-E Nikon microscope.

### Data analysis

Data was analysed in GraphPad Prism v10. Data shown is presented as means ± SD (*Figure 1*) or ± SEM (*Figures 2 and 3*), except for violin plots, where median and quartiles are presented. Two-tailed unpaired Welch's t-test (Gaussian-distributed data) or Mann-Whitney test (non-Gaussian-distributed data) were used for statistical analysis of data with only two experimental groups. For analysis of data with three or more experimental groups, One-way or Two-way ANOVA were performed. When datasets did not follow a Gaussian distribution, data was transformed to a logarithmic scale and parametric statistical analysis was performed on transformed datasets (*MacDonald, 2014*). If logarithmic-transformed data still did not follow a Gaussian distribution, untransformed data was analysed by non-parametric tests (standard or multiple Mann-Whitney tests). Statistical significance was set as: ns – not statistically significant, * $p<0.05$; ** $p<0.01$; *** $p<0.001$, **** $p<0.0001$.

## Acknowledgements

We thank the Francis Crick Institute High-Throughput Screening Science Technology Platform (STP), the Flow Cytometry STP, the Biological Research Facility and the Imaging STP for support. We thank Jeanette Wagener and Aïcha Stierlen for their help in experiment performance and setup. We thank Andreas Wack's lab for providing protocols and support. We want to thank all members of the Treeck lab for critical and continuous discussion. This work was supported by an award to MT from the Wellcome Trust (223192/Z/21/Z), and the Francis Crick Institute, which receives its core funding from Cancer Research UK (CC2132), the UK Medical Research Council (CC2132), and the Wellcome Trust (CC2132). The Science Technology Platforms at the Francis Crick Institute, which receive funding from Cancer Research UK (CC0199), the UK Medical Research Council (CC0199), and the Wellcome Trust (CC0199). FT is supported by the Deutsche Forschungsgemeinschaft (TO 1349/1-1). JCY is funded by an MRC Career Development award (MR/V03314X/1). We thank VEuPathDB (*Amos et al., 2022*) for providing access to the Toxoplasma databases. The funders play no role in study design, data collection and analysis, decision to publish, or preparation of the manuscript.

## Additional information

#### Competing interests

Moritz Treeck: Reviewing editor, *eLife*. The other authors declare that no competing interests exist.

## Funding

| Funder | Grant reference number | Author |
|---|---|---|
| Francis Crick Institute | CC2132 | Francesca Torelli<br>Simon W Butterworth<br>Joanna C Young<br>Moritz Treeck |
| Wellcome Trust | 10.35802/223192 | Francesca Torelli<br>Diogo M da Fonseca<br>Moritz Treeck |
| Deutsche Forschungsgemeinschaft | TO 1349/1-1 | Francesca Torelli |
| Medical Research Council | MR/V03314X/1 | Joanna C Young |

The funders had no role in study design, data collection and interpretation, or the decision to submit the work for publication. For the purpose of Open Access, the authors have applied a CC BY public copyright license to any Author Accepted Manuscript version arising from this submission.

## Author contributions

Francesca Torelli, Conceptualization, Formal analysis, Investigation, Visualization, Methodology, Writing – original draft, Writing – review and editing; Diogo M da Fonseca, Conceptualization, Validation, Investigation, Visualization, Methodology, Writing – original draft, Writing – review and editing; Simon W Butterworth, Investigation, Methodology, Writing – review and editing; Joanna C Young, Conceptualization, Investigation, Writing – review and editing; Moritz Treeck, Conceptualization, Supervision, Funding acquisition, Writing – original draft, Project administration, Writing – review and editing

## Author ORCIDs

Francesca Torelli ⓘ https://orcid.org/0000-0002-2299-9543
Diogo M da Fonseca ⓘ http://orcid.org/0000-0002-2796-6572
Moritz Treeck ⓘ https://orcid.org/0000-0002-9727-6657

## Ethics

C57BL/6J (Jackson Laboratories, RRID:IMSR_JAX:000664), C57BL/6NTac (RRID:IMSR_TAC:B6) and Rag2 N12 C57BL/6N (Rag2-deficient; Rag2$^{-/-}$, RRID:IMSR_TAC:RAGN12; Taconic) mice were bred and housed under specific pathogen-free conditions in the biological research facility at the Francis Crick Institute. Mice maintenance and handling adhered to the Home Office UK Animals Scientific Procedures Act 1986. All work and procedures performed were approved by the UK Home Office and performed in accordance with the granted Project License (P1A20E3F9), the Francis Crick Institute Ethical Review Panel, and conforms to European Union directive 2010/63/EU.

Reviewer #1 (Public review): https://doi.org/10.7554/eLife.102592.3.sa1
Reviewer #2 (Public review): https://doi.org/10.7554/eLife.102592.3.sa2
Author response https://doi.org/10.7554/eLife.102592.3.sa3

# Additional files

## Supplementary files
- MDAR checklist
- Supplementary file 1. Primers used in this study.

## Data availability

All data generated or analysed during this study are included in the manuscript and supporting files; source data files have been provided for all figures.

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
