## [Editor Report · eLife Assessment]

This **important** study shows that *Toxoplasma gondii* uses paracrine mechanisms, in addition to cell-intrinsic methods, to evade the host immune system, with MYR1 playing a key role in transporting effector molecules into host cells. The authors present **convincing** evidence that in vivo, MYR1-deficient parasites can be rescued by wild-type parasites, revealing a limitation in pooled CRISPR screens, where such paracrine effects may obscure the identification of key parasite pathways involved in immune evasion

---

## [Referee Report · Reviewer #1 (Public review)]

Previous studies have highlighted some of these paracrine activities of Toxoplasma - and Rasogi et al (mBio, 2020) used a single cell sequencing approach of cells infected in vitro with the WT or MYR KO parasites - and one of their conclusions was that MYR-1 dependent paracrine activities counteract ROP-dependent processes. Similarly, Chen et al (JEM 2020) highlighted that a particular rhoptry protein (ROP16) could be injected into uninfected macrophages and move them to an anti-inflammatory state that might benefit the parasite.

Caveats around immunity and as yet no insight into how this works. In Fig 2 there is a marked defect in the ability of the parasites to expand at day 2 and day 5. Together, these data sets suggest that this paracrine effect mediated by MYR-1 works early - well before the development of adaptive responses.

Comments on revisions:

The authors have provided their perspective on the original review. There were some previous comments that revolved around whether some of the early changes were masked by pooling data sets where they have reiterated that it is not statistically different. Would have been nice to have seen out addressed by having experiments that were appropriately powered. But it's their call.

---

## [Referee Report · Reviewer #2 (Public review)]

Summary:

In this manuscript by Torelli et al., the authors propose that the major function of MYR1 and MYR1-dependent secreted proteins is to contribute to parasite survival in a paracrine manner rather than to protect parasites from cell-autonomous immune response. The authors conclude that these paracrine effects rescue ∆MYR1 or knockouts of MYR1-dependent effectors within pooled in vivo CRISPR screens.

Strengths:

The authors raised a more general concern that pooled CRISPR screens (not only in Toxoplasma but also other microbes or cancers) would miss important genes by "paracrine masking effect". Although there is no doubt that pooled CRISPR screens (especially in vivo CRISPR screens) are powerful techniques, I think this topic could be of interest to those fields and researchers.

Weaknesses:

In this version, the reviewer is not entirely convinced of the 'paracrine masking effect' because the in vivo experiments should include appropriate controls (see major point 2) in the first submission.

After the revision, although no experiments were added, this reviewer considered that the points have been sufficiently discussed and commented on.

---

## [Author Response]

The following is the authors’ response to the original reviews.

We thank the reviewers for their time and thoughtful comments on our manuscript.

We realised a preliminary version of Figure 2 was initially submitted, which we are replacing now with a novel version. Differences between the two figures are : (1) The schematic in Figure 2a was replaced with a new one in line with that of Figure 3a; (2) in Figure 2c details about the statistical analysis were removed from the legend and one datapoint that was erroneously removed at day 5 for the ΔMYR1-Luc condition was included. Regardless, these changes do not affect the results and the conclusions initially drawn.

**Public Reviews:**

**Reviewer #1 (Public review):**
Previous studies have highlighted some of these paracrine activities of Toxoplasma - and Rasogi et al (mBio, 2020) used a single cell sequencing approach of cells infected in vitro with the WT or MYR KO parasites - and one of their conclusions was that MYR-1 dependent paracrine activities counteract ROP-dependent processes.Similarly, Chen et al (JEM 2020) highlighted that a particular rhoptry protein (ROP16) could be injected into uninfected macrophages and move them to an anti-inflammatory state that might benefit the parasite.

We are aware of both these studies, where the injection of rhoptry proteins into cells that the parasite does not invade alters the host transcriptional profile establishing a permissive environment. However, here we propose a different paracrine effect that goes beyond the injected/uninfected cell. Specifically, we propose that one or more MYR1-dependent effectors alter the cytokine secretion profile of infected cells, which leads to overall changes in the immune response such as cell types recruited to the site of infection, or the activation state.

There are caveats around immunity and as yet no insight into how this works. In Figure 2 there is a marked defect in the ability of the parasites to expand at day 2 and day 5. Together, these data sets suggest that this paracrine effect mediated by MYR-1 works early - well before the development of adaptive responses.

Yes, we also hypothesise an early effect based on the data. Growth continues until day 5 at least, and then plateaus towards day 7, which makes us believe that the effect takes place within the first 5 days. We agree with the reviewer that the MYR1-mediated rescue acts before the involvement of the adaptive immune response, which is supported by our results obtained in Rag2-/- mice shown in Figure 3e.

**Reviewer #2 (Public review):**
Summary:In this manuscript by Torelli et al., the authors propose that the major function of MYR1 and MYR1-dependent secreted proteins is to contribute to parasite survival in a paracrine manner rather than to protect parasites from cell-autonomous immune response. The authors conclude that these paracrine effects rescue ∆MYR1 or knockouts of MYR1-dependent effectors within pooled in vivo CRISPR screens.Strengths:The authors raised a more general concern that pooled CRISPR screens (not only in Toxoplasma but also other microbes or cancers) would miss important genes by "paracrine masking effect". Although there is no doubt that pooled CRISPR screens (especially in vivo CRISPR screens) are powerful techniques, I think this topic could be of interest to those fields and researchers.Weaknesses:In this version, the reviewer is not entirely convinced of the 'paracrine masking effect' because the in vivo experiments should include appropriate controls (see major point 2).(1) It is convincing that co-infection of WT and ∆MYR1 parasites could rescue the growth of ∆MYR1 in mice shown by in vivo luciferase imaging. Also, this is consistent with ∆MYR1 parasites showing no in vivo fitness defect in the in vivo CRISPR screens conducted by several groups. Meanwhile, it has been reported previously and shown in this manuscript that ∆MYR1 parasites have an in vitro growth defect; however, ∆MYR1 parasites show no in vitro fitness defect the in vitro pooled CRISPR screen. The authors show that the competition defect of ∆MYR1 parasites cannot be rescued by co-infection with WT parasites in Figure 1c, which might indicate that no paracrine rescue occurred in an in vitro environment. The authors seem not to mention these discrepancies between in vitro CRISPR screens and in vitro competition assays. Why do ∆MYR1 parasites possess neutral in vitro fitness scores in in vitro CRISPR screens? Could the authors describe a reasonable hypothesis?

The reviewer raises a very interesting point, which at this stage, we cannot fully explain. A technical explanation could be that the relatively small growth defect detected for clean KOs, is not well represented in the CRISPR screens due to the variability of guides, where smaller differences in growth are not reliably captured and hidden within the noise of the assays. Another technical explanation may be median-centering: if the majority of KOs in the pool have a small growth defect, median centering would push these towards a zero. We have observed and reported this phenomenon in Young et al., 2019 for libraries containing a larger fraction of genes with a negative fitness score. In the library used here focusing on secreted proteins, we have not observed a strong trend to negative fitness scores, but cannot exclude smaller shifts. Because we have no solid base to favour any of the above mentioned explanations, we have decided to not speculate too much on this in the manuscript. However, we wanted to show all the data as the difference between these results may not be technical, but biological, which could inform future studies or results by us and others.

(2) The authors developed a mixed infection assay with an inoculum containing a 20:80 ratio of ΔMYR1-Luc parasites with either WT parasites or ΔMYR1 mutants not expressing luciferase, showing that the in vivo growth defect of ∆MYR1 parasites is rescued by the presence of WT parasites. Since this experiment lacks appropriate controls, interpretation could be difficult. Is this phenomenon specific to MYR1? If a co-inoculum of ∆GRA12-Luc with either WT parasites or GRA12 parasites not expressing luciferase is included, the data could be appropriately interpreted.

We are not quite sure what appropriate controls the reviewer refers to. We show here in Figures 3c and 3f that increasing parasite load by co-infecting mice with ∆MYR1 parasites is not sufficient to rescue ∆MYR1-Luc parasite growth. Co-infection with WT parasites, however, does result in increased ∆MYR1-Luc parasitaemia at day 7 p.i., indicating that MYR1 competence is required for the *in vivo* trans-rescue we describe. As ∆GRA12 parasites have a very strong cell-autonomous restriction *in vitro* and severe growth defect *in vivo* (Torelli et al., BioRxiv), these parasites would be rapidly depleted, which is also observed in all CRISPR screens from various laboratories. Therefore we do not think that co-infection with GRA12-deficient parasites would be an informative experiment here. We do speculate that mutant parasites for other proteins required for export (i.e. MYR 2, 3, 4, ROP17) could also be trans-rescued in addition to mutants for other MYR-dependent proteins such as GRA24 and GRA28, which remodel cytokine secretion and could individually, or synergistically, affect host cell immunity. Dissecting which *Toxoplasma* factor/s and host cytokine signalling pathways drive this trans-rescue effect is highly interesting, but beyond the scope of this manuscript. Here, we focused on the basic concept that an individual mutant can be rescued in trans *in vivo,* which we think is of importance beyond the field of *Toxoplasma* research.

(3) In the Discussion part, the authors argue that the rescue phenotype of mixed infection is not due to co-infection of host cells (lines 307-310). This data is important to support the authors' paracrine hypothesis and should be shown in the main figure.

We understand the reviewer’s concern for rescue by co-infection of the same cell, but we largely exclude this hypothesis as *Toxoplasma* cell-autonomous effectors, such as GRA12 and ROP18, would also be rescued if that were to happen on a larger scale. We previously performed an *in vivo* experiment to assess co-infection rates of peritoneal exudate cells (PECs) by imaging using infection doses comparable to those used in the trans-rescue experiments. The total infection rate of PECs was 2.3%, so the overall number of infected cells per image was low, and not suitable for publication purposes. We tried to capture more cells using FACS analysis, however, PECs are highly autofluorescent in the yellow/green channels, which prevented us from drawing adequate conclusions using our GFP and mCherry strains. Because we see no rescue of GRA12 or ROP18 in CRISPR screens, and the overall *in vivo* co-infection rates were very low as observed by imaging, we did not think that generating strains expressing different fluorochromes compatible with standard FACS analysis, and then performing more *in vivo* experiments was best use of resources at the time.

(4) In the Discussion part, the authors assume that the rescue phenotype is the result of multiple MYR1-dependent effectors. I admit that this hypothesis could be possible since a recently published paper described the concerted action of numerous MYR1-dependent or independent effectors contributing to the hypermigration of infected cells (Ten Hoeve et al., mBio, 2024). I think this paragraph would be kind of overstated since the authors did not test any of the candidate effectors. Since the authors possess ∆IST parasites, they can test whether IST is involved in the "paracrine masking effect" or not to support their claim.

MYR1 deletion impairs the export of multiple *Toxoplasma* effectors into the host cell, including GRA16, GRA24, GRA28, HCE1/TEEGR etc, many of which can influence cytokine levels. As such, we speculate that it is a combination of multiple effector proteins that are responsible for the trans-rescue. As stated above, which parasite effectors, host cell types and cytokines are involved in the phenotype we describe are part of ongoing and future studies. Here, we wanted to focus on the key message, that in *in vivo* CRISPR screens, paracrine rescue of individual mutants can occur. While we will test IST mutants, it is probably not the top candidate as it only prevents upregulation of ISGs after exposure to IFN-γ, but has probably no role in already stimulated cells. As we still observe strong rescue past day 3, when IFN-γ levels are already elevated (Nishiyama 2020 Parasitol Int), IST probably plays no dominant role.

**Recommendations for the authors:**

**Reviewer #1 (Recommendations for the authors):**
(1) Figure 1 - it's not obvious what concentration of IFN-gamma is being used in these assays (sorry if this is stated somewhere else).

All *in vitro* experiments were performed with 100 U/ml IFN-γ as stated in the Material & Methods section, however added this information in the figure legend of Figure 1.

(2) Figure 3 This reviewer wonders if earlier differences are buried in the data sets. In Figure 3b it looks like there are early differences but this is lost in the collated data analysis in 3c. An early difference is quite apparent in Figure 2.

We agree with the reviewer that a difference is visible at day 3 and 5 in Figure 3b, however differences between experimental groups became statistically significant only at day 7 in Figure 3c (N = 4 biological replicates). We cannot compare results between Figure 3c and Figure 2c as the latter reports 100% WT or ΔMYR1 infections and not 20:80 mixes.

(3) The authors conclude from their in vitro studies that MYR-1 is not required for in vitro growth in IFN-g activated macrophages. Given that the WT parasites still rescue MYR KO parasites in RAG mice it does imply that this paracrine effect would impact early innate responses. Since RAG mice do have a strong ILC/NK cell response that leads to the local production of IFN-g it would seem like a reasonable candidate. Do the authors know if the MYR KO have improved growth in the absence of IFN-g in vivo? This could be done using KO mice or with IFN-g neutralization.

MYR1 displayed a neutral score in CRISPR screens in IFN-γ KO mice (Tachibana et al Cell Reports 2023), suggesting that lack of IFN-γ does not specifically improve MYR1 mutant growth compared to other mutants in a pool. We believe that the rescue is rather driven by other cytokines that have been shown to be altered in a MYR1 dependent manner (i.e CCL2, IL-6, IL-12). But as laid out before, this is subject of future studies.

This is a submission that might benefit from a graphical model of how the authors view this system working.

We agree with the reviewer and we added a graphical model to the manuscript.

**Reviewer #2 (Recommendations for the authors):**
The authors previously published a study that combines CRISPR screens in Toxoplasma and host transcriptome by scRNA-seq (Butterworth et al., Cell Host Microbe 2023). I think the authors possess transcriptome of ∆MYR1-infected HFFs. Although I understand this screen is conducted in in-vitro culture and human fibroblasts, are there any differentially expressed genes or pathways that could explain the paracrine rescue phenomenon described in this manuscript?

We thank the reviewer for this insightful comment, which is however hard to address. Thousands of host cell genes within multiple pathways are affected by MYR1 deletion (Naor et al. mBio 2018; Butterworth et al. Cell Host Microbe 2023). Therefore the PerturbSeq dataset is not helpful to pinpoint specific immune mechanisms of rescue, and is speculative without any experimentation to back it up. However, we added a sentence in line 350 of the discussion to highlight known MYR1-related effects on immune-related pathways. “Individual MYR-related effectors that may be responsible for the paracrine rescue have not been investigated here and we hypothesise that the phenotype is likely the concerted result of multiple effectors that affect cytokine secretion. For example, previous studies showed that both GRA18 and GRA28 can induce release of CCL22 from infected cells (He 2018 eLife; Rudzki 2021 mBio), while GRA16 and HCE1/TEEGR impair NF-kB signalling and the potential release of pro-inflammatory cytokines such as IL-6, IL-1β and TNF (Seo 2020 Int J Mol Sci; Braun 2019 Nat Microbiol). Regardless of the effector(s), our results highlight an important novel function of MYR1-dependent effectors by establishing a supportive environment in *trans* for *Toxoplasma* growth within the peritoneum.”